# ON THE GEOMETRY OF ANALOGICAL REASONING IN LATENT SPACE*

**Oleg Dats**
Department of Computer Science
Ukrainian Catholic University
Lviv, Ukraine
`dats.pn@ucu.edu.ua`

## ABSTRACT

Analogical reasoning is the ability to infer and apply a consistent relation across examples. This ability imposes a simple geometric constraint: example pairs should differ by a shared displacement in representation space. Yet most neural architectures do not enforce this structure explicitly, instead relying on implicit attention-based mechanisms. We study analogical reasoning from a geometric perspective and show that it can be realized by a minimal latent-space translation structure. We instantiate this principle with an Encoder–Reasoner–Decoder (ERD) architecture, where each example pair defines a latent difference vector and a task-level transformation is obtained by averaging these differences, corresponding to the least-squares solution of a shared linear relation. Using the Abstraction and Reasoning Corpus (ARC) as a diagnostic benchmark, we show that explicitly enforcing this geometric constraint improves generalization relative to implicit attention-based reasoning approaches, while reducing computation from quadratic to linear in the number of examples. The learned representations exhibit clear geometric structure: difference vectors cluster by task and form consistent parallelogram relations, providing direct evidence of explicit analogical geometry. These results suggest that analogical reasoning does not require complex symbolic machinery, large-scale attention, or auxiliary training losses, but can emerge from enforcing a simple and interpretable geometric structure in latent space.

## 1 INTRODUCTION

**Analogical reasoning.** The ability to infer and apply a shared relational structure across different instances is widely regarded as a core component of abstraction and generalization in human cognition (Gentner, 1983; Lake et al., 2017). This view has motivated the development of diagnostic benchmarks that aim to isolate analogical reasoning independently of scale and prior knowledge. The Abstraction and Reasoning Corpus (ARC) was explicitly proposed to test abstraction via analogy, emphasizing transformation inference over pattern recognition (Chollet, 2019). Performance on ARC remains challenging for contemporary machine learning systems, highlighting analogical reasoning as a distinct and unsolved capability rather than a byproduct of data scale or heuristic pattern matching (Marcus, 2018; Bahdanau et al., 2018).

**Implicit reasoning in modern neural networks.** Modern neural architectures, particularly transformer-based models, can solve analogy-like tasks (Brown et al., 2020), but the underlying reasoning is typically implemented implicitly. Relations are distributed across attention weights, token interactions, or extended inference procedures, rather than represented as explicit operations (Elhage et al., 2021). As a result, it is difficult to isolate the mechanism responsible for analogical reasoning, analyze its behavior, or reuse it as a modular component. This implicitness is also coupled with quadratic scaling in the number of examples, as relational structure is modeled through pairwise interactions rather than task-level abstractions (Vaswani et al., 2017).

---

*Code available at `https://github.com/odats/arc_erd`

**Contributions.** We investigate whether analogical reasoning can be realized as an explicit geometric operation in latent space, and whether enforcing this structure improves efficiency, interpretability, and generalization compared to implicit attention-based reasoning. Our contributions are as follows:

- **Analogical Reasoning as an Explicit Geometric Primitive.** We formulate analogical reasoning as estimating a single task-level translation in latent space that aligns all example pairs. This principle is instantiated by an Encoder–Reasoner–Decoder (ERD) architecture that implements the transformation explicitly, rather than distributing it implicitly across attention.

- **Linear Complexity from Minimal Geometric Structure.** By operating on a single latent operator per task, ERD achieves linear $O(n)$ memory and compute during training and constant $O(1)$ memory at inference, where $n$ is the number of example pairs, in contrast to the quadratic $O(n^2)$ scaling of attention-based autoregressive decoders.

- **Interpretable Latent Geometry.** The inferred transformation vectors exhibit coherent geometric structure, including tight within-task clustering and consistent parallelogram relations, indicating that analogical reasoning is realized as an explicit additive operator in latent space.

- **Improved Generalization with Reduced Compute.** On the ARC benchmark, enforcing a shared latent operator improves generalization under both standard and test-time training. ERD consistently solves approximately twice as many tasks as a GPT-style baseline trained under comparable conditions, achieves up to a 40% relative improvement with test-time adaptation, and requires roughly $4\times$ less compute due to linear scaling.

## 2 RELATED WORK

**Analogical reasoning in neural networks.** Neural approaches to analogical reasoning are largely dominated by transformer-based models that treat analogy as an emergent property of attention, autoregressive decoding, and test-time computation, with ARC serving as a prominent diagnostic setting. Strong performance can be achieved by amplifying this implicit reasoning through extensive solution sampling and selection, without architectural changes (Greenblatt, 2024). Complementary analyses highlight different aspects of neural reasoning: Li et al. (2024a) show that inductive and transductive paradigms succeed on different subsets of tasks, while Cole et al. (2025) demonstrate that training on large, diverse math and logic datasets substantially improves generalization. Test-time training (TTT) further enhances abstract reasoning by adapting model parameters on a per-task basis (Akyürek et al., 2024). Other work emphasizes the role of inductive biases in visual reasoning; Li et al. (2024b) show that explicit 2D positional structure, object-centric representations, and connectivity priors significantly improve performance. Additional methods such as chain-of-thought prompting (Wei et al., 2022) and RL-based objectives like R1 (Guo et al., 2025) can be viewed as extensions of implicit GPT-style reasoning that increase internal computation and test-time exploration, strengthening the same paradigm rather than altering its inductive bias.

**Geometric inductive biases.** A broad line of work in geometry-grounded representation learning shows that enforcing structural constraints such as symmetry, equivariance, or metric structure can improve generalization, interpretability, and efficiency. Group-equivariant models explicitly encode symmetry assumptions into the architecture (Cohen & Welling, 2016), while work on geometric embeddings demonstrates that the choice of latent-space geometry itself strongly shapes representation quality and generalization (Nickel & Kiela, 2017). More broadly, geometric deep learning provides a unifying framework for incorporating such structural priors across domains (Bronstein et al., 2021). From this perspective, architectural inductive biases encode assumptions about the structure of the problem domain directly into the model. Our work aligns with this view by treating analogical reasoning itself as a geometric constraint on latent representations, rather than as an emergent property of scale or training procedure.

**Latent-space algebra.** Latent-space arithmetic has long been observed as an emergent phenomenon in representation learning, most notably in Word2Vec-style embeddings where semantic relations correspond to linear offsets (Mikolov et al., 2013). Similar observations have been made in

disentangled representation learning and vision models, where latent dimensions admit interpretable manipulations (Higgins et al., 2017; Sitzmann et al., 2020). Explicit additive relation modeling has also been explored in knowledge-graph embeddings such as TransE (Bordes et al., 2013), where relations are represented as translations between entity embeddings. Recent work has applied similar vector arithmetic to ARC by learning visual embeddings and combining them via addition and subtraction at inference time (Thoms et al., 2023), demonstrating that latent offsets can partially capture task structure. In contrast, our approach does not rely on post hoc vector arithmetic over pretrained embeddings, but enforces a single task-specific latent transformation as an explicit architectural operation, ensuring that all examples within a task satisfy a shared geometric constraint. As a result, the latent algebra functions as a reasoning primitive rather than a representational artifact.

## 3 METHODS

### 3.1 PROBLEM SETTING AND GEOMETRIC VIEW OF ANALOGY

We consider analogical reasoning problems of the form $A : B :: C :?$, instantiated in ARC as a set of example input–output pairs $\{(a_i, b_i)\}_{i=1}^n$ and a query input $a_{\text{task}}$. Each grid is embedded independently by an encoder into a sequence of latent vectors

$$\mathbf{a}_i, \mathbf{b}_i \in \mathbb{R}^{S \times D},$$

where $S$ denotes the number of grid tokens and $D$ the embedding dimension ($S = 1024$, $D = 256$ in all experiments).

We adopt a geometric perspective in which each task is characterized by a single shared latent translation that maps all inputs to their corresponding outputs. Analogical reasoning is thus formulated as estimating and applying this translation in latent space.

### 3.2 LATENT ANALOGY AS A GEOMETRIC OPERATOR

For each example pair, we define a latent difference vector by averaging token-wise differences:

$$\boldsymbol{\delta}_i = \frac{1}{S} \sum_{s=1}^{S} \left( \mathbf{b}_i^{(s)} - \mathbf{a}_i^{(s)} \right) \in \mathbb{R}^D,$$

where $\mathbf{a}_i^{(s)}$ and $\mathbf{b}_i^{(s)}$ denote the $s$-th token embeddings.

Each $\boldsymbol{\delta}_i$ can be viewed as a noisy observation of an underlying task-level translation. We estimate the task translation by

$$\hat{x} = \frac{1}{n} \sum_{i=1}^{n} \boldsymbol{\delta}_i.$$

This estimator is the closed-form solution to the least-squares problem

$$\hat{x} = \arg \min_{\mathbf{x} \in \mathbb{R}^D} \sum_{i=1}^{n} \sum_{s=1}^{S} \left\| \left( \mathbf{a}_i^{(s)} + \mathbf{x} \right) - \mathbf{b}_i^{(s)} \right\|_2^2.$$

Geometrically, $\hat{x}$ corresponds to the centroid of the token-wise difference vectors in latent space, i.e., a projection onto a shared task direction. Analogical reasoning in ERD is not learned via gradient descent but implemented as a geometric least-squares estimator in latent space.

### 3.3 ENCODER–REASONER–DECODER ARCHITECTURE

We implement this geometric reasoning primitive using an Encoder–Reasoner–Decoder (ERD) architecture, shown in Figure 1.

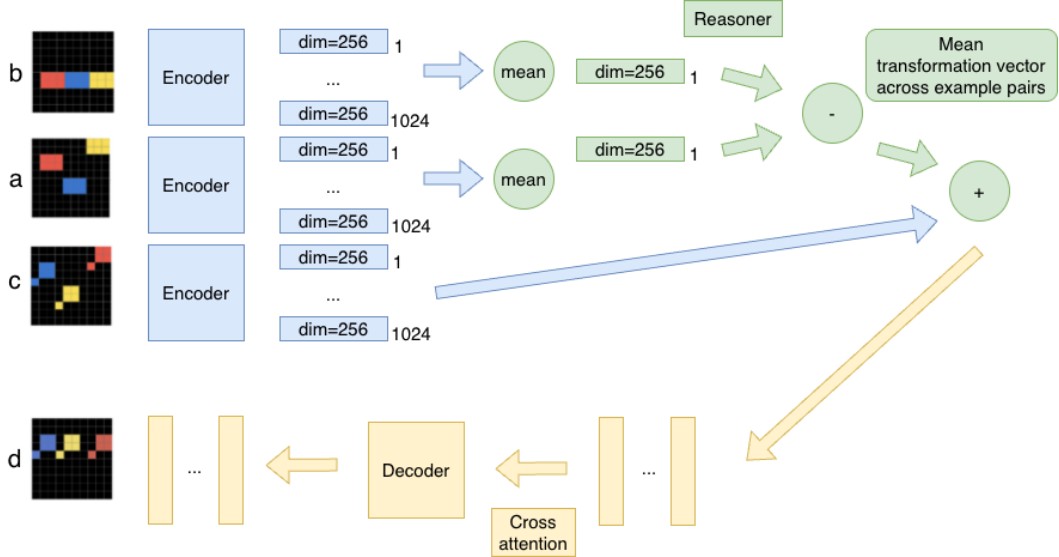

Figure 1: Encoder–Reasoner–Decoder (ERD) architecture.

**Encoder.** Each ARC grid is flattened into a sequence of 1024 discrete tokens and embedded independently using a BERT-like transformer (Devlin et al., 2019), without pretraining. The encoder maps each grid into a latent representation in $\mathbb{R}^{1024 \times 256}$. It performs no cross-example interaction and no task-level aggregation; its role is to define a latent space in which analogical relations correspond to simple geometric structure.

**Reasoner.** The reasoner applies the previously defined latent translation. For each example pair, it computes the mean token-wise displacement $\boldsymbol{\delta}_i$ and aggregates these across examples to obtain a single task-level translation $\hat{x}$. This translation is broadcast and added uniformly to all token embeddings of the task input, producing a translated latent representation that approximates the task output. The same geometric operation is applied to all tokens, enforcing a shared translation across the grid.

**Decoder.** The decoder reconstructs the output grid from the translated latent representation in $\mathbb{R}^{1024 \times 256}$. It is a GPT-style autoregressive model that accesses the translated latent vectors via cross-attention indexed by grid position and outputs, for each of the 1024 positions, a categorical distribution over output tokens. The decoder has no access to example pairs and performs no analogical reasoning; it simply maps translated latent vectors back to the discrete grid domain. All relational structure is imposed by the explicit geometric translation applied by the reasoner.

Enforcing analogical reasoning as a shared latent translation induces a geometry in which grids can be embedded independently and in parallel. All example pairs are condensed into a single task-level latent representation, resulting in linear scaling rather than the quadratic scaling of GPT-style attention.

### 3.4   TRAINING OBJECTIVE

Training enforces analogical reasoning only in the forward direction. The model is trained to decode transformed input embeddings into output grids by minimizing pixel-wise cross-entropy:

$$\mathcal{L} = \mathrm{CE}(\mathrm{decode}(\mathbf{a}_i + \hat{x}),\ b_i) + \mathrm{CE}(\mathrm{decode}(\mathbf{a}_{\mathrm{task}} + \hat{x}),\ b_{\mathrm{task}}) .$$

No inverse reconstruction or auxiliary losses are used; all learning signal flows through a single explicit latent-space translation.

### 3.5 COMPUTATIONAL COMPLEXITY

Let $n$ denote the number of example pairs in a task and $s$ the number of tokens per grid.

**GPT-style models.** GPT-style decoders represent relations implicitly through pairwise attention over all tokens from all examples, the task input, and the output. This results in quadratic self-attention cost

$$\big((2n+2)s\big)^2 = O(n^2)$$

so both compute and memory scale quadratically with the number of examples, reflecting the need to model all pairwise interactions explicitly.

**ERD.** ERD replaces pairwise token interactions with a single explicit geometric operator in latent space. Each grid is encoded independently, yielding

$$(2n+1)s = O(n)$$

encoder cost. The reasoner computes a task-level transformation via simple averaging of latent difference vectors, a constant-cost geometric aggregation. The decoder attends only to the transformed task embedding and output tokens, resulting in attention cost independent of $n$.

During training, ERD stores per-example embeddings, requiring $O(n)$ memory. At inference time, only the estimated task-level transformation is retained, yielding $O(1)$ memory. Thus, enforcing analogical reasoning as an explicit geometric structure reduces both compute and memory from quadratic to linear scaling.

## 4 RESULTS

We evaluate whether enforcing analogical reasoning as an explicit latent-space operator induces coherent geometric structure, improves generalization, and reduces computational cost, rather than optimizing absolute ARC performance.

### 4.1 ARC AS A GEOMETRIC ANALOGY BENCHMARK

Each ARC task instantiates analogies of the form $A : B :: C :?$ via multiple input–output examples and a query input. From a geometric perspective, each task implicitly defines a shared latent transformation mapping inputs to outputs. ARC therefore serves as a diagnostic benchmark for evaluating whether a model learns a coherent task-level operator in representation space, rather than encoding relations implicitly through attention.

### 4.2 IMPLEMENTATION DETAILS

**Dataset.** All experiments are conducted on the RE-ARC dataset (Hodel, 2024) for training and the original ARC benchmark for evaluation, with no overlap in tasks. Each ARC task consists of multiple input–output example pairs and a single query input. Grids are padded to $32 \times 32$ and flattened into sequences of 1024 tokens, where each token corresponds to a discrete color value in $\{0, \ldots, 9\}$. Standard ARC augmentations are applied, including rotations, reflections, and color permutations.

**Evaluation.** For each ARC task, we perform 256 independent evaluation attempts by randomly sampling augmented subsets of the available demonstration examples. Each attempt produces a full model prediction and is counted as correct if the predicted output grid exactly matches the ground truth. We report the empirical probability of solving each task, defined as the fraction of correct attempts across the 256 samples.

**Models.** We evaluate the proposed Encoder–Reasoner–Decoder (ERD) architecture against a GPT-style autoregressive baseline trained from scratch. The baseline is a standard GPT-style decoder with absolute positional embeddings, trained using a next-token prediction objective with cross-entropy loss. Both ERD and the GPT baseline use 256-dimensional embeddings and 4 transformer layers, and 8 attention heads.

**Training.** All models are trained from scratch on the RE-ARC 1000 verified generated examples for each of the 400 training tasks using the AdamW optimizer with learning rate $3 \times 10^{-4}$ and linear warmup. Training is performed for up to 100k epochs. To scale training, we use multi-GPU data parallelism: the effective batch size is scaled linearly with the number of GPUs, and the learning rate is scaled proportionally following standard linear scaling rules. For test-time training (TTT), models are further fine-tuned on a single unseen task for two epochs using 256 augmented samples drawn from the task's demonstration examples, using the same optimizer and learning-rate schedule.

**Compute setup.** All experiments are conducted on NVIDIA RTX 4090 GPUs. Due to quadratic scaling in the number of examples, the GPT-style baseline requires substantially more hardware to match the effective batch size and training throughput of ERD. Under compute-aligned conditions, ERD achieves approximately a $4\times$ reduction in total compute, consistent with its linear scaling compared to the quadratic scaling of attention-based decoders.

## 4.3 FINAL PERFORMANCE AND GENERALIZATION ON ARC

Table 1 reports the empirical probability of solving ARC tasks under standard training and test-time training (TTT).

| Model | Standard | + TTT |
|---|---|---|
| GPT | 3% | 19% |
| ERD | 6% | 26% |
| ERD (512d, 12L) | 8% | 33% |

Table 1: ARC task success rates with and without test-time training (TTT).

Across all settings, ERD consistently outperforms the GPT-style baseline while requiring substantially less total compute due to linear scaling in the number of examples. Results with a larger ERD variant (512d, 12L) show that these gains persist and further improve with model scaling.

## 4.4 QUANTITATIVE EVALUATION OF LATENT TRANSFORMATIONS

ERD represents each ARC task by a single latent transformation vector $\hat{x}$, estimated as the mean of per-example differences $b_i - a_i$ (Section 3.1). If this representation is coherent, per-example transformations should align closely with $\hat{x}$ within a task and remain unaligned across tasks.

We quantify this behavior using cosine similarity, measuring directional alignment, and mean squared error (MSE), measuring dispersion around $\hat{x}$. Table 2 reports results for same-task and different-task comparisons on both training and test splits.

| Split | Cos (same) | Cos (diff) | MSE (same) | MSE (diff) | $\|\hat{x}\|$ |
|---|---|---|---|---|---|
| ARC (train) | 0.92 | 0.01 | 0.0052 | 0.15 | 3.84 |
| ARC (test) | 0.91 | 0.01 | 0.0061 | 0.14 | 3.71 |

Table 2: Quantitative alignment of per-example transformations $b_i - a_i$ with the estimated task-level operator $\hat{x}$.

Same-task cosine similarity is consistently high, while different-task similarity remains near zero. Likewise, same-task MSE is an order of magnitude lower than different-task MSE, indicating tight clustering around the task-level operator. The stable norm of $\hat{x}$ across splits suggests consistent operator scale and generalization to unseen tasks.

## 4.5 VISUALIZATION OF EMERGENT LATENT ANALOGICAL STRUCTURE

ERD enforces analogical reasoning by solving the latent relation $a_i + \hat{x} \approx b_i$. A necessary consequence is that per-example difference vectors $b_i - a_i$ concentrate around a single task-specific

direction, forming an approximately affine structure in latent space. This structure is not enforced by any explicit geometric loss, but arises solely from the architectural constraint.

To visualize this effect, we project example-level difference vectors into two dimensions using principal component analysis (PCA), which approximately preserves affine relations under linear projection.

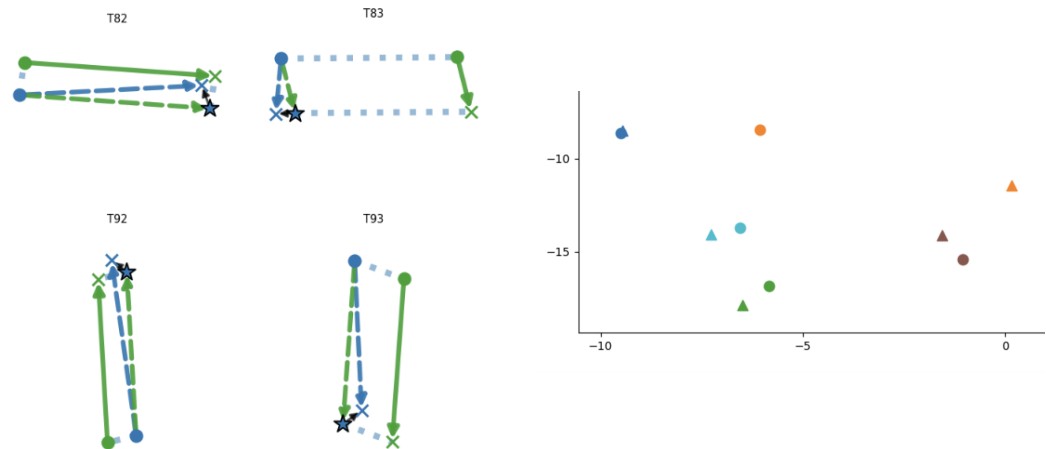

Figure 2: Emergent latent analogical geometry. All visualized tasks are randomly sampled from the ARC training set. **(a)** Parallelogram relations $B - A \approx D - C$ within ARC tasks. Blue dashed arrows denote ground-truth input–output transformations. Green solid arrows denote the true transformation from the example input to the example output. Green dashed arrows denote the transformation predicted by applying the estimated task-level operator $\hat{x}$. For visualization clarity, each panel uses a single example pair. Stars indicate predicted outputs and crosses indicate ground-truth outputs. **(b)** Clustering of difference vectors $b_i - a_i$ by task in a 2D PCA projection, despite no task supervision; points with the same color correspond to the same task.

Figure 2a shows parallelogram structure within tasks, a hallmark of additive relational encoding and shared task-level transformations. Figure 2b shows tight within-task clustering and clear inter-task separation, directly justifying estimation of $\hat{x}$ via simple averaging.

### 4.6    ABLATION

We evaluate the impact of key geometric and structural choices underlying ERD. All results use only the image reconstruction loss; auxiliary latent losses, including embedding reconstruction loss, identity reconstruction loss, transformation norm regularization, and contrastive-style objectives, yield comparable performance and are therefore unnecessary. We additionally replace simple aggregation of example-level differences with more expressive alternatives, such as learned token-wise weighting and Transformer-based scoring, but observe no measurable improvement, indicating that increased estimator complexity does not benefit analogical generalization.

| Model Variant | Regular | TTT |
|---|---|---|
| Token-wise transformation + 1D positional encoding | 3% | 15.5% |
| + Global mean transformation | 5% | 24.5% |
| + 2D positional encoding (**ERD**) | **6%** | **26%** |

Table 3: Progressive ablation of architectural components. Starting from a token-wise transformation model with 1D positional encoding, introducing a global task-level transformation substantially improves performance. Adding 2D positional encoding further improves results by aligning latent representations with the spatial structure of ARC grids.

The global transformation corresponds to estimating a single task-level latent operator shared across all tokens and examples, rather than allowing independent per-token transformations. Introducing this shared operator produces the largest performance gain, suggesting that ARC tasks are better modeled by a coherent task-level analogical relation rather than independent token-wise mappings.

Two-dimensional positional encoding provides an additional improvement by preserving spatial structure in the latent representation. Since ARC tasks are defined over 2D grids, incorporating explicit 2D positional structure aligns the representation space with the geometry of the underlying domain, further stabilizing the estimated transformation.

## 5 CONCLUSION

We showed that analogical reasoning can be realized as a minimal and explicit geometric operation in latent space, rather than as an emergent property of large-scale attention or auxiliary training procedures. By enforcing a shared latent translation across example pairs, the proposed Encoder–Reasoner–Decoder architecture induces coherent task-level structure, yielding improved generalization, linear computational scaling, and directly interpretable latent geometry on ARC. The emergence of consistent parallelogram relations and tight within-task clustering demonstrates that enforcing a simple geometric constraint is sufficient to support analogical abstraction without additional losses or symbolic machinery. These results suggest that analogical reasoning admits a compact geometric formulation and highlight explicit latent operators as a promising direction for building efficient, interpretable, and scalable reasoning systems.

### 5.1 LIMITATIONS AND FUTURE DIRECTIONS

Algebraic composition of transformation vectors and disentanglement is underexplored. Discrete latent spaces, such as those induced by VQ-VAE (van den Oord et al., 2017), may enable richer, compositional transformations. Another limitation is the lack of a straightforward way to incorporate pretrained models. A promising direction is to explore architectures like T5-Gemma (Zhang et al., 2025) that decouple encoder and decoder modules, potentially allowing ERD's encoder and reasoner to be combined with a pretrained decoder. Finally, vector-based transformations have two structural issues: (1) handling constant outputs, since $a_1 + \hat{x} = b$ and $a_2 + \hat{x} = b$ together imply $a_1 = a_2$; and (2) fixed capacity—some transformations require far more information than a single vector can represent (e.g., empty $+ \hat{x} = a$, where the vector $\hat{x}$ cannot encode an entire complex output). A natural future direction is to model transformations as matrices or higher-order operators rather than single vectors.

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
