# OpenReview forum: "On the Geometry of Analogical Reasoning in Latent Space"
_ICLR.cc/2026/Workshop/GRaM — ICLR 2026 Workshop GRaM Poster_

### Official Review · Reviewer_NAsG · 2026-02-15
**On the Geometry of Analogical Reasoning in Latent Space**

**Rating:** 7
**Confidence:** 4

**Review:**

This paper studies analogical reasoning from a geometric perspective and proposes that analogy can be implemented as a shared latent-space translation across example pairs. The authors introduce an Encoder–Reasoner–Decoder (ERD) architecture in which each input–output pair defines a difference vector, and the task-level transformation is estimated via least-squares averaging. This explicit geometric operator replaces attention-based relational modeling, yielding linear complexity in the number of examples. On ARC, ERD outperforms a GPT-style baseline under comparable training conditions while requiring substantially less compute. The learned latent representations exhibit clear clustering and parallelogram structure consistent with additive relational geometry.

Strengths:
- The formulation of analogical reasoning as a shared latent translation is elegant and theoretically well-motivated.
- The architecture enforces a geometric constraint rather than relying on implicit attention mechanisms, improving interpretability.
- ERD achieves linear scaling in the number of examples, reducing compute relative to quadratic attention-based baselines.
- The clustering of different vectors and parallelogram structure (Section 4.5) provides compelling qualitative and quantitative evidence that the intended geometric structure emerges.
- The reasoning module is minimal and analytically grounded (least-squares estimator), avoiding auxiliary losses or complex training schemes.

Weaknesses:
- Absolute ARC success rates remain low (6–8% standard, 27–33% with TTT), so while relative gains over a small GPT baseline are clear, the approach remains far from state-of-the-art ARC performance.
- Enforcing a single global translation may limit expressivity; some ARC transformations require non-uniform or compositional operators that cannot be captured by a single additive vector (acknowledged in Section 5.1).
- The GPT-style baseline is trained from scratch and relatively small; comparisons against stronger pretrained or structured ARC solvers would provide clearer positioning.
- The method may implicitly favour tasks that admit approximate affine transformations in latent space, potentially limiting generality beyond ARC-style grid transformations.
- While the manuscript is concise (9 pages), substantial space is devoted to conceptual motivation and visualisation. A deeper theoretical analysis (limits of translation-only operators, compositional extensions) or broader empirical evaluation could strengthen the contribution.

**Pmlr Suitability:**

Yes

---

### Official Review · Reviewer_V5Hx · 2026-02-17
**Review of On the Geometry of Analogical Reasoning in Latent Space**

**Rating:** 7
**Confidence:** 4

**Review:**

The paper studies learning analogical reasoning. This is commonly seen as a central aspect of intelligence, useful for example when applying abstract rules in novel situations. The authors focus on the ARC challenge in their theoretical setting and experiments, which is reasonable and makes the paper easy to follow.

Analogical reasoning capabilities sometimes emerges implicitly in neural architectures, but they are hard to isolate. This paper proposes modelling analogical reasoning as explicit translation in latent space. Each task is represented as a single latent translation, which is estimated from examples by averaging the latent offsets. To solve the test case, the model adds the inferred translation and decodes the output. Training uses a simple pixel-wise cross-entropy loss on the decoded outputs.

The paper fits the theme of the workshop.

**strengths**

- The approach is elegant and simple
- The method is interpretable: each task is represented by a single translation vector
- There is experimental evidence that the learned translations are non-trivial and task-specific (instead of the decoder simply overfitting)
- Inference is efficient: solving a task only requires estimating the translation and applying it to the test input tokens

**weaknesses**

- Some more examples would have been interesting to see what kinds of transformations this approach actually handles. Figure 1 shows a task where all objects move in the same direction in the grid, which is arguably the simplest case for this kind of approach. What if shapes do not move in the same direction in the grid, but the translation is for example towards a center object. Can the latent space express the right coordinate system so that the relation still looks like a global translation?
- ARC tasks of course include transformations that are not captured by a single translation. It's not clear how to combine the geometric objective with richer or composite transformations.

**Pmlr Suitability:**

Yes

---

### Official Review · Reviewer_7Rii · 2026-02-25
**Review of On the Geometry of Analogical Reasoning in Latent Space**

**Rating:** 6
**Confidence:** 3

**Review:**

This paper proposes that analogical reasoning can be formalized as an explicit latent-space translation. Given example pairs $\{(a_i, b_i)\}_{i=1}^n$, a task-level transformation is estimated as the mean difference vector $\hat{x} = \frac{1}{n}\sum_i \delta_i$, where $\delta_i = \frac{1}{S}\sum_s (b_i^{(s)} - a_i^{(s)})$, corresponding to the least-squares solution of a shared linear relation. This is instantiated in an
Encoder--Reasoner--Decoder (ERD) architecture and evaluated on ARC, where it outperforms a GPT-style baseline trained under comparable conditions while reducing compute from $O(n^2)$ to $O(n)$.

Strengths

- The geometric framing is elegant and principled, with a clean least-squares analytical foundation.
- Same-task cosine similarity of 0.92 versus near-zero cross-task similarity emerges solely from the architectural constraint, without any auxiliary geometric loss.
- Replacing pairwise attention with a single aggregated vector yields linear rather than quadratic scaling, with an empirically verified ~4x compute reduction.
- The ablation cleanly isolates the global translation constraint as responsible for the majority of the performance gain.

Weaknesses

- The GPT baseline is only 4 layers and 256 dimensions, making it hard to attribute ERD's gains to the geometric inductive bias rather than the baseline simply being under-resourced. A parameter-matched comparison would strengthen the claim.
- Absolute ARC success rates remain low (6-8% standard, 27-33% TTT), and the paper lacks any comparison against structured or object-centric baselines beyond this small GPT model.
- The single-vector formulation is a fundamental expressivity ceiling; tasks with spatially non-uniform or information-asymmetric transformations are structurally unsolvable, and the paper does not characterize what fraction of ARC tasks this affects.
- Confidence intervals are absent from Table 1, which is important given the stochastic evaluation procedure.

**Pmlr Suitability:**

Yes

---

### Meta-Review · Area_Chair_Yecz · 2026-02-27

**Decision:**

Accept

**Metareview:**

Reviewers found the geometric approach to analogical reasoning to be elegant.  There were some concerns with the strength of the baselines and the absolute empirical performance.

**Relevance To Proceedings:**

Yes — suitable for PMLR (long paper)

**Relevance To Workshop:**

Yes — suitable for GRaM

---

### Decision · Program_Chairs · 2026-03-02

Accept (Poster)